# Fungal Taxa Responsible for Mucormycosis/“Black Fungus” among COVID-19 Patients in India

**DOI:** 10.3390/jof7080641

**Published:** 2021-08-07

**Authors:** Pradeep K. Divakar

**Affiliations:** Department of Pharmacology, Pharmacognosy and Botany, Faculty of Pharmacy, Complutense University of Madrid, 28040 Madrid, Spain; pdivakar@farm.ucm.es; Tel.: +34-913-941-994; Fax: +34-913-941-774

**Keywords:** black fungus, mucromycosis, COVID-19 outbreak, India, metagenomics

## Abstract

Mucormycosis is caused by fungi belonging to the order Mucorales. The term “Black Fungus” has been widely applied to human pathogenic Mucorales in India. They mainly infect the sinuses and brain, lungs, stomach and intestines, and skin. While this has been considered a rare disease, thousands of cases have been reported during the second wave of COVID-19 in India, between the months of April and June 2021. Hitherto, more than 45,374 cases and over 4300 deaths have been reported among COVID-19 patients across India from April 2021 to July 21, 2021. Though the mortality rate is estimated to be 50%, it could be above 90% if left untreated. In India, *Rhizopus arrhizus* has been related to be the most common species to cause human mucormycosis, followed by *Apophysomyces variabilis*, *Rhizopus microsporus*, and *R. homothallicus*. Accurate sample identification of human pathogenic Mucorales species is challenging especially due to the frequent lack of diagnostic morphological features. Traditionally, the culture-based approach has been extensively used to isolate and characterize human pathogenic Mucorales. However, this may not be an appropriate approach to objectively isolate and characterize all species, as the germination and growth of fungal spores are highly dependent on culture media and environmental conditions. Therefore, a robust approach to the accurate and rapid identification of human pathogenic Mucorales species is a prerequisite. The metagenomic approach comprehensively sequences and analyzes all genetic material in a complex biological sample and, consequently, this could be an appropriate approach to objectively characterize human pathogenic Mucorales taxa without the need for in vitro culture. The precise identification of the species will not only be useful for the correct diagnosis of this disease, but also for the development of antifungal drugs specific for each species. Accurate and rapid species identification is desperately needed to save lives in the mucormycosis outbreak among COVID-19 patients in India and neighboring countries.

## Main Text

Mucormycosis, or the so-called “Black Fungus” in India, is caused by members of the filamentous mold fungus, the Mucorales, such as *Apophysomyces*, *Cokeromyces*, *Cunninghamella*, *Lichtheimia*, *Mortierella*, *Mucor*, *Rhizopus*, *Rhizomucor*, *Saksenaea*, *Syncephalastrum*, and *Thamnostylum* [1,2]. The genera of Mucorales are one of the best decomposers of organic materials and are often found in decaying organic materials such as rooted fruits and vegetables, plant litter, and animal manure. They reproduce rapidly through asexual spores that develop endogenously within a vesicle called a sporangium [2]. The term “Black Fungus” is applied to human pathogenic Mucorales species due to the formation of black-colored sporangium. In general, the fungal spores including Mucorales are airborne and found in indoor and outdoor air, and in dust. Mucorales have also been reported in hospital environments such as hospital bed sheets, negative-pressure rooms, water leaks, contaminated medical equipment, and building works [3,4]. In May 2021, these have also been linked to mechanical ventilation in the intensive care unit and unsterilized medical oxygen cylinder tubes in different hospitals across India, however, the doctors’ opinion on this matter is divided [5,6]. They mostly infect the sinuses and brain (rhinocerebral mucormycosis), lungs (pulmonary mucormycosis), stomach and intestines (gastrointestinal mucormycosis), and skin (cutaneous mucormycosis) [7]. Mucormycosis is usually spread by inhaling sporangiospores, or by eating contaminated food or receiving spores in an open wound. It should be noted that it is not transmitted between people. The sporangiospores of Mucorales range from 3 to 11 μm in diameter and are easily aerosolized [8]. These are mainly dispersed in the indoor and outdoor environment by the wind. While sporangiospore acquisition varies and depends on the causative species found in the environment, the respiratory tract and skin are the main entry points [8].

Mucormycosis has often been linked to natural disasters, such as the 2004 Indian Ocean tsunami and the 2011 Missouri tornado. However, between the months of April and May 2021, thousands of cases have been reported during the second wave of SARS-CoV-2 (hereinafter COVID-19) in India. So far, more than 45,374 cases and 4300 deaths have been reported among COVID-19 patients across India during the second wave of COVID-19 (from April 2021 to 21 July 2021) [9,10]. Maharashtra province was hit hard by this disease and reported above 9000 cases and 1000 deaths followed by Gujrat with 5500 cases and 785 deaths [11,12]. Andhra Pradesh was the third-worst affected province, with 2303 reported cases and 157 deaths [13]. Hundreds of cases have also been reported in Bihar, Chandigarh, Chhattisgarh, Delhi, Goa, Haryana, Karnataka, Kerala, Madhya Pradesh, Punjab, Rajasthan, Tamil Nadu, Telangana, Uttar Pradesh, and Uttarakhand. Mucormycosis cases among COVID-19 patients, including death toll, are continuously increasing in India. Moreover, cases are also being reported in neighboring countries such as Bangladesh, Nepal, and Pakistan. Furthermore, Bangladesh has already reported a death on May 25 allied with “Black Fungus”. Pakistan has also reported five cases of mucormycosis in May 2021 and four had died as of 27 May 2021 [14]. In India, this disease has often been linked to diabetic patients or immuno-compromised people or those who received high doses of steroids for COVID-19 treatments [15,16]. However, it has also been reported in some young people without a history of immunosuppression [16]. According to the International Diabetes Federation (IDF), the highest proportion of the population with diabetes is registered in the United States, followed by Pakistan, China, Brazil, India, and Bangladesh, though mucormycosis cases in the United States, Pakistan, China, Brazil, and Bangladesh were far fewer compared to India [15]. It should be noted that undiagnosed diabetes in India is relatively high (around 57%) (IDF, 2019); and thus, undiagnosed or uncontrolled diabetes can be responsible for a large number of mucormycosis in India. Nonetheless, this needs a thorough investigation. It is worth noting that during the first wave of COVID-19 in India, only 44 cases and nine deaths were reported across India in mid-December 2020 [17]. It is estimated that India has 70–80-fold higher disease burden comparing any country in the world prior to COVID-19 [18]. Highlighting this, one third of the 45,374 cases are non-COVID-19-related (i.e., above 13,000 cases in a 7–8 weeks), but are associated with uncontrolled and poorly controlled diabetes mellitus or immuno-compromised individuals [19]. India has the second-largest number of adults aged 20–79 years with diabetes mellitus [20]. Moreover, a high population in India do not have regular testing of blood sugar levels especially due to a remarkable overload of patients in the hospitals [21,22]. While no single factor has been associated yet with human mucormycosis, it is tending that the uncontrolled and poorly controlled diabetes mellitus may be responsible for emerging epidemic of mucormycosis in India. Moreover, a higher load of sporangiospores of Mucorales in the indoor and outdoor air in India is most likely especially due to tropical and humid climate environment [21]. Therefore, it should be noted that the epidemic emergence of post COVID-19 mucormycosis in India is most likely and the country must be prepared for that a priori. According to the US Centers for Disease Control and Prevention (CDC), this is a rare disease with less than 1% infection rate worldwide with 50% mortality [23]. However, the death rate in India is relatively high and could be above 90% if left untreated. Indeed, this is not a rare disease in India, as the country declared mucormycosis an epidemic in all 29 provinces, in the last week of May 2021 [24]. Some doctors have voiced their opinion on this deadly disease in India to be considered as a pandemic within a pandemic.

Accurate sample identification of the species causing mucormycosis is challenging, especially due to the lack of a robust approach to treat, as well as the frequent lack of diagnostic morphological features [25,26]. Traditionally, the culture-based approach has been used to isolate and characterize human pathogenic Mucorales [18]. As the germination and growth of fungal spores are highly dependent on the culture media and environmental conditions, not all spores are able to grow in a culture medium. Thus, it hampers to isolate all species responsible for causing human mucormycosis. So far, about 27 species have been identified and associated with human mucormycosis [2]. *Rhizopus arrhizus* was the most common species to cause human mucormycosis in India, followed by *Apophysomyces variabilis*, *Rhizopus microsporus*, and *R. homothallicus* [2,18]. Other less common species were *Apophysomyces elegans*, *Lichtheimia ramosa*, *Mucor irregularis*, *Rhizomucor pusillus*, *Saksenaea erythrospora*, *Syncephalastrum racemosum*, and *Thamnostylum lucknowense*. Recently, matrix-assisted laser desorption ionization time-of-flight mass spectrometry (MALDI-TOF MS) has been proposed for consistent and fast identification of human pathogenic Mucorales at the species level [27]. However, this may not be a feasible approach to the identification of filamentous fungi, including human pathogenic Mucorales, remarkably due to the frequent occurrence of morphologically cryptic species. For example, three cryptic species have so far been detected in the monospecific fungal genus *Apophysomyces* [26]. This is the second most common taxa to cause human mucormycosis in India, additionally, because only 1% of the total estimated fungal species (i.e., 3 to 5 or 12 million, [28]) has been described so far. In those cases, a DNA sequence-based species identification is the most reliable method. The internal transcribed spacer region (ITS), is the universal DNA barcode for fungi and is commonly used for the identification of fungal species, including Mucorales [29]. Moreover, it is also recommended for species identification of human pathogenic Mucorales [25]. Recently, several PCR methods have been developed for the diagnosis of human mucormycosis, such as nested PCR, real-time PCR (qPCR), and nested PCR combined with RFLP [30]. Multiplex real-time qPCR targeting the ITS1/ITS2 region for *R. oryzae*, *R. microsporus*, and *Mucor* spp. and different regions of 18S and 28S rDNA for Mucorales have also been developed [31,32]. Further, a real-time qPCR commercial kit (Mucorgenius^®^, PathoNostics, Maastricht, The Netherlands) is also available for fast diagnostic test from blood samples of patients [33]. These approaches could be applied targeting ITS1 or ITS 2 regions for a quick clinical diagnosis of mucormycosis. However, the conventional PCR including multiplex RT PCR approach for targeted species may not be suitable to detect low fungal burden infection [34], and to identify all human pathogenic Mucorales taxa and accurate species identification in complex biological mixtures including environmental samples. Thus, a comprehensive approach is needed. Metagenomics has been shown to be a robust approach to species identification in complex biological mixtures including environmental and gut samples [35]. Recently, this approach has also been applied to assess the diversity of airborne fungi in the hospital environment [36]. The metagenomic approach comprehensively sequences and analyzes all genetic material in a complex biological sample. Total DNA could be easily isolated from infected tissue (without the need for culture) and metagenome sequencing using targeted amplicon sequencing: For example, the internal transcribed spacer region (ITS) could be performed with next-generation sequencing (NGS). Since the spores are found in the indoor air of the hospital building, they can be easily captured, e.g., with Hirst spore trap [37], which has been used for over 60 years in aerobiological sampling. Total DNA could be isolated and metagenome sequencing performed from these spores. The taxonomic assignment of these sequences will make it possible to alert about the occurrence of human pathogenic Mucorales species in hospitals and a priori precautions could be taken. Additionally, metagenomics is being used for the identification and prediction of airborne biological particles, including fungal spores, in the outdoor environment. 

Mucormycosis treatments involves a combination of surgical intervention of infected tissues and antifungal therapy [25,38]. The removal of prejudicing factors for infection, such as hyperglycemia, metabolic acidosis, deferoxamine administration, immunosuppressive drugs, and neutropenia, is also crucial. Intravenous (IV) and lipid formulation of amphotericin B is highly recommended for initial therapy [25]. For patients who have responded to a lipid formulation of amphotericin B, posaconazole or isavuconazole could be used for oral step-down therapy. Intravenous (IV) posaconazole or isavuconazole is used as salvage therapy for patients who did not respond to or could not tolerate amphotericin B. Isavuconazole or posaconazole may be administered as maintenance therapy [25]. Moreover, combination of amphotericin B and echinocandin or amphotericin B and azoles has also been administrated to treat invasive mucormycosis [25,39]. Although combination therapy is not recommended in the major treatment guidelines, detailed studies are needed to establish whether or not combination therapy is beneficial. Mucormycosis remains a therapeutic challenge in India and lipid formulation of amphotericin B was widely administered in conjunction with surgical interventions to treat the infection. In several patients, the eyes and or upper jaw were removed to stop the infection and save lives [16]. So far, only three potential antifungal drugs viz. amphotericin B, posaconazole, and isavuconazole are available for efficient treatment of infection. While amphotericin B and posaconazole were the most effective antifungal drugs, species–specific differences were reported in an in vitro study [40]. This indicates that accurate species identification is crucial not only for the correct diagnosis of the disease, but also for developing species-specific antifungal drugs. Therefore, accurate and rapid species identification is desperately needed to save lives in the mucormycosis outbreak among COVID-19 patients in India. 

## Data Availability

Not applicable.

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
