# Peer review of "Fungal Taxa Responsible for Mucormycosis/“Black Fungus” among COVID-19 Patients in India"

_jof, 2021, doi:10.3390/jof7080641_

Round 1
Reviewer 1 Report
I agree with the authors that it is important to draw the attention of scientist and lay people to mucormycosis and how devastating the disease is, especially for an opinion article. The edits in response the reviewers are adequate but I would fix the following: Abstract Line 13: make sure that the 4 in 40,845 is not deleted. Introduction Line 51: change "the doctor's opinion is divided" to "the doctors' opinion on this matter is divided". Another important issue is the lack of reference to the high burden of the disease in India prior to COVID-19. It is estimated that India has 70-80 fold higher disease burden vs. any country in the world prior to COVID-19 (Chander et al J. Fungi 2018, 4, 46; doi:10.3390/jof4020046). Highlighting this, one third of the 40,845 cases are non-COVID-19-related (this is >13,000 cases in a 7-8 weeks) (Ravani et al. Indian J Ophthalmol 2021;69:1563-8). This needs to be mentioned in the report I would also delete lines 86-90, since the disease prior to COVID was not a reportable disease and the numbers listed by the authors doesn't reflect the real disease burden in India prior to COVID-19.Author Response
Reviewer 1
I agree with the authors that it is important to draw the attention of scientist and lay people to mucormycosis and how devastating the disease is, especially for an opinion article.
Response: I highly appreciate for your valuable comments and suggestions and that have been very helpful to me to improving the manuscript.
The edits in response the reviewers are adequate but I would fix the following: Abstract Line 13: make sure that the 4 in 40,845 is not deleted.
Response: Thanks for pointing out that. The case numbers have been double checked. Moreover, I have updated the case numbers and death tools till July 21. These are the latest numbers announced by Indian health minister so far.
Modified text: “Hitherto, more than 45,374 cases and over 4,300 deaths have been reported among COVID-19 patients across India from April 2021 to July 21, 2021.”
Introduction Line 51: change "the doctor's opinion is divided" to "the doctors' opinion on this matter is divided".
Response: Done
Another important issue is the lack of reference to the high burden of the disease in India prior to COVID-19. It is estimated that India has 70-80-fold higher disease burden vs. any country in the world prior to COVID-19 (Chander et al J. Fungi 2018, 4, 46; doi:10.3390/jof4020046). Highlighting this, one third of the 40,845 cases are non-COVID-19-related (this is >13,000 cases in a 7-8 weeks) (Ravani et al. Indian J Ophthalmol 2021;69:1563-8). This needs to be mentioned in the report I would also delete lines 86-90, since the disease prior to COVID was not a reportable disease and the numbers listed by the authors doesn't reflect the real disease burden in India prior to COVID-19.
Response: Thanks for this wonderful suggestion. I have included the suggested sentence. Moreover, lines 86-90 has been deleted. Following sentences has been added in the manuscript.
New text: “It is estimated that India has 70-80-fold higher disease burden comparing any country in the world prior to COVID-19 [18]. Highlighting this, one third of the 45,374 cases are non-COVID-19-related (i.e. above 13,000 cases in a 7-8 weeks), however, associated with uncontrolled and poorly controlled diabetes mellitus or immuno-compromised individuals [19].”
Reviewer 2 Report
Dear Authors,
The manuscript ID: jof-1320143 entitled “Fungal taxa responsible for Mucormycosis / “Black Fungus” among COVID-19 patients in India” written by Pradeep K. Divakar is very current. The incidence of mucormycosis increased significantly during the COVID-19 pandemic in India, which is very worrying. Mucormycosis is a rare but potentially fatal infection if inadequately treated. This opinion is interesting to readers and scientists.
I have some suggestions in order to improve paper, which are the following:
- „Introduction” – the manuscript contains one section - "Introduction"?
- Line 117: „mucromycosis” – mucormycosis;
- More information about mucormycosis, “black fungus”, mucormycosis cases and emergence of post-COVID-19 mucormycosis should be add. The following references may be helpful:
- https://www.thelancet.com/action/showPdf?pii=S2213-2600%2821%2900265-4
- John TM, Jacob CN, Kontoyiannis DP. When Uncontrolled Diabetes Mellitus and Severe COVID-19 Converge: The Perfect Storm for Mucormycosis. J Fungi (Basel). 2021 Apr 15;7(4):298. doi: 10.3390/jof7040298. PMID: 33920755; PMCID: PMC8071133.
- Singh P. Black fungus: here is a list of states with highest number of mucormycosis cases. Hindustan Times, May 21, 2021; https://www.hindustantimes.com/india-news/black-fungus-states-with-highest-number-of-mucormycosis-cases-101621559394002.html
- Gupta, Amod; Sharma, Aman, Chakrabarti, Arunaloke. The emergence of post-COVID-19 mucormycosis in India. Can we prevent it? Indian Journal of Ophthalmology: July 2021, Volume 69, Issue 7, p 1645-1647. doi: 10.4103/ijo.IJO_1392_21
According to me, this manuscript is very topical, valuable and may be accepted for the publication in “Journal of Fungi”, after major review.
With highest regards,

Author Response
Reviewer 2
The manuscript ID: jof-1320143 entitled “Fungal taxa responsible for Mucormycosis / “Black Fungus” among COVID-19 patients in India” written by Pradeep K. Divakar is very current. The incidence of mucormycosis increased significantly during the COVID-19 pandemic in India, which is very worrying. Mucormycosis is a rare but potentially fatal infection if inadequately treated.This opinion is interesting to readers and scientists.
Response: I thank you very much for your feedback and wonderful suggestions that have been very helpful for me. Accordingly, I have modified the text.
I have some suggestions in order to improve paper, which are the following:
1. „Introduction” – the manuscript contains one section - "Introduction"?
Response: I think this is a journal format. But in any case, I have replaced it with a heading “Main text”
2. Line 117: „mucromycosis” – mucormycosis;
Response: Done
3. More information about mucormycosis, “black fungus”, mucormycosis cases and emergence of post-COVID-19 mucormycosis should be add. The following references may be helpful:
- https://www.thelancet.com/action/showPdf?pii=S2213-2600%2821%2900265-4
- John TM, Jacob CN, Kontoyiannis DP. When Uncontrolled Diabetes Mellitus and Severe COVID-19 Converge: The Perfect Storm for Mucormycosis. J Fungi (Basel). 2021 Apr 15;7(4):298. doi: 10.3390/jof7040298. PMID: 33920755; PMCID: PMC8071133.
- Singh P. Black fungus: here is a list of states with highest number of mucormycosis cases. Hindustan Times, May 21, 2021; https://www.hindustantimes.com/india-news/black-fungus-states-with-highest-number-of-mucormycosis-cases-101621559394002.html
- Gupta, Amod; Sharma, Aman, Chakrabarti, Arunaloke. The emergence of post-COVID-19 mucormycosis in India. Can we prevent it? Indian Journal of Ophthalmology: July 2021, Volume 69, Issue 7, p 1645-1647. doi: 10.4103/ijo.IJO_1392_21
Response: Thanks for this wonderful suggestion. Accordingly, I have added a short paragraph and cited the suggested references. These have been very helpful. The following paragraph has been added.
New text: “It is estimated that India has 70-80-fold higher disease burden comparing any country in the world prior to COVID-19 [18]. Highlighting this, one third of the 45,374 cases are non-COVID-19-related (i.e. above 13,000 cases in a 7-8 weeks), however, associated with uncontrolled and poorly controlled diabetes mellitus or immuno-compromised individuals [19]. India has the second-largest number of adults aged 20–79 years with diabetes mellitus [20]. Moreover, a high population in India do not have regular testing of blood sugar levels especially due to a remarkable overload of patients in the hospitals [21-22]. While no single factor has been associated yet with human mucormycosis, it is tending that the uncontrolled and poorly controlled diabetes mellitus may be responsible for emerging epidemic of mucormycosis in India. Moreover, a higher load of sporangiospores of Mucorales in the indoor and outdoor air in India is most likely especially due to tropical and humid climate environment [21]. Therefore, it should be noted that the epidemic emergence of post COVID-19 mucromycosis in India is most likely and the country must be prepared for that a priori.”
According to me, this manuscript is very topical, valuable and may be accepted for the publication in “Journal of Fungi”, after major review.
With highest regards,
This manuscript is a resubmission of an earlier submission. The following is a list of the peer review reports and author responses from that submission.
Round 1
Reviewer 1 Report
This is an opinion on the current cases of mucormycosis in India. Although the style is good, the opinion falls short of giving any new data on what is already published and fails to add to the knowledge that we currently have. In fact, this opinion doesn’t have updated numbers reported by the government of India to be >31 k cases of mucormycosis as of June 13 (#mucormycosis).
Update the number of cases on line 13. It is now surpassed 31,000 cases and many more death.
Italicize all organisms’ names in the entire document.
On line 36, replace the word species with genera.
The linking of the current outbreak in India to contaminated mechanical ventilation/unsterilized oxygen cylinders should be referenced since I am not aware of a published study on this topic.
There are contradictory statements such as the one on line 114 stating that Apophysomyces is responsible for 60% of the documented cases of human mucormycosis in India when they clearly state that Rhizopus is the major cause of infection followed by Apophysomyces.
Line 117 correct relabel to reliable.
The use of metagenomics and NGS is not a realistic diagnostic assay for individuals in India. What do we know about PCR-based assays? Can they be used? There are many studies used different PCR-assay for diagnosis of mucormycosis.
There are recommendations published recently for treatment and caring for mucormycosis. These are not discussed nor referenced.
Reviewer 2 Report
Reviewer’s report
The manuscript entitled “Fungal taxa responsible for Mucormycosis / “Black Fungus” 2 among COVID-19 patients in India.“ by Pradeep K. Divakar. Mucorymcosis is a life-threating infection in immunocompromised patients caused by certain member of the fungal order Mucorales. During the SARS-CoV-2 pandemic the number of pations with mucormycosis significantly increased in India. The main species are Apophysomyces variabilis, Rhizopus microsporus, and R. homothallicus. The author has collected the available datas of cases from India.
The manuscript is clearly written, but there are some issues the authors should address by making modification to the manuscript before accepting.
Minor points:
- Author should check the manuscript, because the name of species and genera must be written in italic.
- Line 8. “Mucromycosis is caused by the filamentous mold fungus” change it. For example: Mucormycosis is caused by fungi belonging to the order Mucorales.
- Line 25. in vitro instead of in-vitro.
- Line 41-44, the sentence must be deleted.
- Line 38. Reference is missing.
- Line 49. Reference is missing
- Line 55. Reference is missing
- Line 66. [9-11]
- Line 97. Reference is missing.
- Line 102. Reference is missing
- Line 163. Date is not bold